# Addressing Challenges in Targeted Therapy for Metastatic Colorectal Cancer

**DOI:** 10.3390/cancers17071098

**Published:** 2025-03-25

**Authors:** Maria El Hage, Zhaoran Su, Michael Linnebacher

**Affiliations:** Molecular Oncology and Immunotherapy, Clinic of General Surgery, Rostock University Medical Center, 18057 Rostock, Germany; maria.elhage11@gmail.com (M.E.H.); zhaoran.su@med.uni-rostock.de (Z.S.)

**Keywords:** mCRC treatment, targeted therapy, biomarkers, combination therapy in metastatic colorectal cancer, primary resistance, tumor heterogeneity, acquired resistance, resistance to antiangiogenic therapy, mode of resistance to EGFR inhibitors, resistance to Her2 inhibition

## Abstract

This review discusses the challenges of treating advanced colorectal cancer that has spread to other parts of the body using targeted therapies, which work by blocking specific pathways that help cancer grow. While therapies that target blood vessel growth exist, proteins like epidermal growth factor receptor, and immune checkpoints, their effectiveness varies widely. Cancer can resist treatment from the beginning due to inter-patient differences as well as differences within a single tumor. Over time, cancer cells adapt and find new ways to grow, making treatment less effective. Combining therapies is often used to overcome resistance, but this approach can increase side effects and treatment costs, making it less accessible for many patients. Further research, especially using genetic testing and blood tests that detect cancer DNA, could help make these treatments more effective and adaptable. This review highlights the complex nature of targeted therapy for advanced colorectal cancer and the need for new strategies to address these challenges.

## 1. Introduction

Over the past two decades, there have been significant breakthroughs in the treatment of metastatic colorectal cancer (mCRC). These advances have been driven primarily by the introduction of new, more effective targeted agents. The optimal approach for the sequential use and combination of these agents is evolving. Treatment decisions are increasingly influenced by a better understanding of tumor biology [1]. The pathogenesis of colorectal cancer (CRC) is shown in Figure 1. Remarkable progress has been made in the development of biomarker-driven targeted therapies for patients with various types of cancer, including melanoma, breast and lung tumors, although precision oncology for patients with CRC continues to lag behind [2].

Several biologic agents have shown activity against mCRC. These drugs specifically target factors and receptors such as vascular endothelial growth factors (VEGFs) and their receptors (VEGFRs), epidermal growth factor receptor (EGFR), human epidermal growth factor receptor 2 (HER2), the BRAFV600E mutation, and immunotherapy with immune checkpoint inhibitors (ICIs), the latter being the latest revolution in tumor treatment [1]. Specific biomarkers have been identified for patients who may benefit from EGFR, HER2, neurotrophin receptor tyrosine kinase (NTRK) fusion and ICI treatments [3]. Biomarkers will be discussed in more detail later.

A general treatment guideline has been established for the use of targeted therapies in mCRC that incorporates the use of biomarkers into clinical practice where the approach to later lines of systemic therapy in mCRC is portrayed in Figure 2:-Among patients diagnosed with mCRC, the presence or absence of RAS mutations is a key determinant in identifying patients who may benefit from EGFR-targeted treatment strategies. In particular, the administration of anti-EGFR monoclonal antibodies (cetuximab, panitumumab) should be limited to patients with RAS wild-type tumors. Conversely, patients with BRAF-mutated tumors generally do not respond to anti-EGFR antibodies as monotherapy. However, in selected cases, the introduction of a BRAF inhibitor has shown some positive responses, although this therapeutic approach is typically reserved for advanced stages of treatment [4].-For individuals with tumors that exhibit deficient mismatch repair (dMMR), typically characterized by high levels of microsatellite instability (MSI-H), initiation of treatment with an immune checkpoint inhibitor should be considered as a first-line therapeutic approach [4].

## 2. Drugs Used in Targeted Therapy for CRC

Primary treatment of cancer refers to the initial therapeutic interventions aimed at eliminating or controlling tumor growth, such as surgery, radiation or chemotherapy. Secondary treatment, also known as adjuvant therapy, follows primary treatment and is designed to reduce the risk of recurrence or treating residual disease. It often includes additional chemotherapy, radiation therapy, targeted therapy, or immunotherapy [5].

We will first discuss the most common biologicals used in the primary and secondary treatment of CRC, where their clinical results are summarized in Table 1.

-
**EGFR inhibitors**


The frequently overexpressed EFGR in CRC is targeted by cetuximab and panitumumab, two monoclonal antibodies approved for the treatment of RAS/BRAF wild-type CRC [6,7].

Cetuximab is approved for the treatment of mCRC in the first and second-line setting in combination with chemotherapy or as a single agent in patients who have failed oxaliplatin or irinotecan-based chemotherapy [6].

Panitumumab is approved as a single agent for the treatment of mCRC after failure of fluoropyrimidine-, oxaliplatin- and irinotecan-based chemotherapy regimens [8,9]. EGFR inhibitors in tumors that are RAS wild-type demonstrate expected response rates of 55–65%, while the actual responses in the clinical settings range from 40–50%. Diminished effectiveness is observed mostly in right-sided tumors because of intrinsic resistance [10,11].

-
**Targeting BRAFV600E**


BRAFV600E-inhibitors are used as specific targeted therapy for BRAFV600E mutated mCRC. This mutation occurs in approximately 8–15% of patients with CRC and is associated with poor prognosis and resistance to standard chemotherapy regimens [1]. The BRAFV600E inhibitors vemurafenib and encorafenib are typically used in the second line setting as part of a combination therapy, often with other targeted therapeutics such as EGFR or MEK inhibitors (e.g., binimetinib), to enhance treatment efficacy [12,13]. Targeting BRAFV600E has achieved moderate success in the clinical setting. Combination therapies seem to improve survival, but fail to achieve the high response rates that are seen in other cancer types (such as Melanoma). The BEACON CRC trial demonstrated that encorafenib + cetuximab ± binimetinib led to improved overall survival (9.0–9.3 months vs. 5.4 months with chemotherapy) and response rates (20–26% vs. 2%). Nevertheless, resistance mechanisms, especially EGFR reactivation, lead to limited effectiveness. BRAFV600E-mutant mCRC remains aggressive, characterized by poor prognosis and an average survival of approximately 12 months [14].

-
**Human epidermal growth factor receptor 2 (HER2) inhibitor**


Trastuzumab and pertuzumab are FDA-approved HER2 inhibitors for the treatment of HER2-positive mCRC. Trastuzumab is approved for use in combination with chemotherapy as a second-line treatment for HER2-positive mCRC. Pertuzumab has been approved by the FDA for dual HER2 blockade treatment in combination with trastuzumab and additional chemotherapy as first-line treatment for HER2-positive mCRC [1,15,16]. Preclinical research indicated that HER2-amplified mCRC is highly responsive to anti-HER2 treatments, especially when employing dual blockade. Nonetheless, clinical studies demonstrated inconsistent effectiveness, where single-agent therapy showed limited advantages, whereas dual blockade resulted in an objective response rate ranging from 9.7% to 35%. Trastuzumab reported the highest efficacy, achieving a 45% objective response rate, although it carries risks of pulmonary toxicity. Although these treatments have not become standard care, they are utilized in certain hospitals and clinical trials, supporting the validity of HER2-targeted approaches in molecularly selected mCRC [17].

-
**MEK inhibitors**


Trametinib and cobimetinib are examples of MEK inhibitors used in clinical practice for the treatment of mCRC. Both drugs are approved for the treatment of melanoma with BRAFV600E or V600L mutations, but are not specifically approved for mCRC [1].

As a result, MEK inhibitors are generally not used in the first-line treatment of mCRC. Instead, they are typically considered in later lines of treatment, particularly for patients with BRAFV600E-mutant or RAS wild-type tumors who have progressed on prior therapies. They are often used in combination with other targeted agents, such as BRAF or EGFR inhibitors, to improve outcomes [13].

-
**Immune checkpoint inhibitors**


Immune checkpoint targeted therapy aims to enhance the immune system’s ability to recognize and control cancer cells by inhibiting the mechanisms used by tumors to evade immune cells, particularly T-cell recognition [18,19]. Currently, ICIs are mostly approved only for highly immunogenic tumor entities such as renal cell cancer, melanoma and NSCLC, with promising clinical results [2]. The PD-1 blockers pembrolizumab and nivolumab, both humanized antibodies, have been approved for the treatment of MSI-H or MMR-deficient solid tumors, including mCRC [9,20]. In particular, treatment in combination with cytotoxic T-lymphocyte antigen 4 inhibitors such as ipilimumab shows benefit in advanced MSI-H or dMMR mCRC [21].

### Challenges in Targeted Therapy

Numerous clinical trials are underway to incorporate additional targeted therapies into the adjuvant setting [22]. As we strive to improve the clinical efficacy of these agents and create curative approaches, complexities and obstacles arise [23]. The challenges facing the use of targeted agents in mCRC are resistance in its many forms, toxic side effects of the drugs that limit both dosing and patient acceptance, and the enormous cost that limits availability to all mCRC patients in need [23].

## 3. Drug Resistance

The results of targeted therapies are variable, with some patients responding completely, some partially, and some not responding at all. Tumors that initially fail to respond are considered to have primary or intrinsic resistance, while those that initially respond may later develop acquired or secondary resistance [23]. The following section reviews the mechanisms underlying these different types of resistance.

### 3.1. Primary Resistance

Tumor heterogeneity is one of the major factors contributing to primary resistance [23].

Heterogeneity is defined when tumor cells within the same tumor site or across different metastatic sites within the same individual exhibit different characteristics, such as histology, metastatic or proliferative capacity, and molecular profiles [24]. It can be divided into inter-patient and intra-tumor heterogeneity, which will be discussed in detail.

CRC is known for its significant heterogeneity in both forms. This is due to the remarkable genomic instability, including a comparatively high percentage of MSI-H, as well as the phenomenon of field cancerization [25,26]. In addition, there is evidence that external factors such as lifestyle, diet, nutrition, environment and microbiome play a role in the development of CRC, affecting not only neoplastic but also non-neoplastic cells, including immune cells, thereby contributing to increased heterogeneity [27,28]. For instance, diets lacking in fruits, vegetables, and fiber, as well as unhealthy lifestyle habits, such as lack of physical activity and smoking have been associated with an increased risk of CRC. Smoking can cause DNA damage by exposing the colon and rectum to carcinogens. In addition, engaging in regular physical exercise has been shown to reduce the risk of CRC by regulating hormone levels, enhancing insulin sensitivity, and improving immune function [29].

Furthermore, the gut microbiome plays a critical role in drug resistance in CRC. The gut microbiome modulates chemotherapy and immune checkpoint inhibitor efficacy through microbial composition, metabolic pathways, and immune interactions [29,30]. In addition, hormonal imbalances, ageing, and stress also contribute to CRC drug resistance by altering tumor biology and immune responses. For instance, hyperinsulinemia and IGF-1 activation stimulate PI3K/AKT/mTOR signaling, whicg in turn enhances tumor proliferation and survival and simultaneously reduces sensitivity to chemotherapy. Cortisol dysregulation due to chronic stress or metabolic dysfunction suppresses immune responses and promotes an inflammatory microenvironment that favors tumor progression [31,32]. Epigenetic modifications accumulate with ageing, which then leads to altering gene expression and reducing the effectiveness of targeted therapies. Immunosenescence, which is characterized by reduced cytotoxic T-cell function and impaired antigen presentation, weakens responses to treatment [33].

In addition, chronic stress activates the hypothalamic-pituitary-adrenal axis, which leads to an increased catecholamine release, hence enhancing adrenergic receptor signaling, leading to increased angiogenesis, tumor invasion, and chemoresistance. This stress-induced immunosuppression weakens anti-tumor immune responses, further compromising treatment efficacy. All the above-mentioned factors collectively create a pro-tumor microenvironment, which increases the likelihood of therapy failure [34].

**a.** 
**Inter-patient heterogeneity**


Tumors exhibit molecular heterogeneity that may be influenced by their origin and location. According to Fearon and Vogelstein’s 1990 model, most sporadic CRCs (approximately 85%) develop from adenomas [35]. Sporadic CRCs are characterized by chromosomal instability, a negative CpG island methylator phenotype (CIMP), microsatellite stability (MSS), mutated KRAS, but wild-type BRAF. On the other hand, approximately 15% of CRCs arise from serrated adenomas and follow a pathway characterized by high CIMP, MSI, and BRAF mutation. A proportion of these tumors have significantly increased mutation rates due to the dMMR genes, resulting in high MSI-H. These tumors represent a minority of CRCs, with a more favorable prognosis in early-stage disease and a lower incidence in advanced stages [36].

Tumor localization in CRC plays a role in molecular and phenotypic variations. Left- and right-sided tumors have different gene expressions and mutational profiles that affect prognosis. Right-sided tumors are associated with a higher prevalence of BRAF mutations, MSI-H, and occur more frequently in patients with genetic predisposition such as Lynch syndrome [37].

In contrast, left-sided tumors show chromosomal instability, and a gene expression pattern associated with the activation of the EGFR pathway. In addition, right-sided tumors had a significantly higher mutation burden compared to left-sided tumors (102 vs. 66 mutations, *p* = 0.004) [38].

Ulivi et al. found different distributions of angiogenic, and inflammatory markers based on tumor localization. Right-sided tumors exhibited higher expression levels of cyclooxygenase-2 (COX2), ephrin type B receptor 4 (EPHB4), and endothelial nitric oxide synthase (eNOS), whereas left-sided tumors had higher baseline inflammatory indices such as neutrophil-to-lymphocyte ratio, platelet-lymphocyte rate, and systemic immune-inflammation index [39]. Finally, microbial abundance and composition, which are thought to contribute to the development of CRC, differ depending on the primary tumor site [40].

These variations result in heterogeneous treatment response and prognosis [41,42].

**b.** 
**Spatial heterogeneity**


Apart from inter-patient heterogeneity, there are three significant forms of spatial heterogeneity in tumorigenesis. First, intra-tumor heterogeneity refers to variations among cells within the same primary tumor. Second, inter-metastatic heterogeneity arises from differences among different metastatic lesions within a single patient. Finally, different mutations may coexist within cells of a single metastatic lesion in the same patient [26]. Figure 3 describes how metastasis can lead to drug resistance. Spatial heterogeneity is influenced by the existence of genetically distinct clones resulting from evolutionary processes, as well as the coexistence of fully differentiated cancer cells and immature cancer stem cells within the same tumor [43]. Kreso et al. demonstrated that functional heterogeneity can exist among tumor cells of the same genetic lineage in CRC, with cells exhibiting different capacities for growth and response to chemotherapy [44]. Furthermore, spatial heterogeneity in CRC is driven by the distribution and activity of immune cells, which are influenced by local tumor cells, stroma, and microenvironmental factors, including the microbiota [18].

Intra-tumor heterogeneity leads to an underestimated view of the mutational landscape obtained from a single needle biopsy, thus affecting treatment precision. Therefore, spatial heterogeneity plays a role in treatment resistance, especially when selective pressure is exerted, such as by late-stage anticancer drugs [45]. This phenomenon is not only caused by the broad mutagenic effect of chemotherapy, but also by the chemotherapy-induced “competitive release” effect, which allows treatment-resistant subclones to repopulate and contribute to tumor relapse [46,47]. In addition, several studies have shown that heterogeneity, as indicated by the presence of multiple subclonal alterations within a tumor, may be associated with adverse outcomes in several cancer types [48].

In CRC patients, Sveen et al. demonstrated that both intra-patient and inter-metastatic heterogeneity significantly influenced prognosis. Patients with low heterogeneity had a three-year PFS rate of 23% and an OS rate of 66%, while those with high heterogeneity had rates of 5% and 18%, respectively [49]. The current lack of reliable standardized methods to assess intra-tumor heterogeneity hinders our ability to fully investigate its clinical implications [50]. To gain a deeper understanding of heterogeneity in primary and metastatic lesions, Wang et al. in a recent review emphasized the importance of investigating not only the genetic, but also the epigenetic, transcriptomic, and proteomic levels [51].

### 3.2. Acquired Resistance

Acquired resistance is inferred when tumor progression occurs after an initial favorable response [23]. In 2010, Jackman et al. proposed a guideline to define acquired resistance to EGFR tyrosine-kinase inhibitors (TKIs) in NSCLC. These concepts can be used as a general reference for the clinical characterization of acquired resistance, with some adaptations. To classify resistance as acquired, patients should meet the following criteria (1) previous treatment with a targetable drug; (2) tumor with an abnormality associated with drug sensitivity and/or clinical benefit from treatment with a targetable drug; (3) systemic disease progression according to Response Evaluation Criteria in Solid Tumors or WHO criteria while on continuous treatment, with no intervening systemic therapy between discontinuation of the targetable drug and initiation of new therapy [52].

Multiple mechanisms leading to acquired resistance have been recognized as shown in Figure 4 [53,54] and can be divided into driver oncogene alterations, abnormal signaling pathway alterations in parallel or downstream fashion, pro-survival signaling through a different or unknown signaling pathway, histologic transformation from one cell lineage to another, and drug tolerant cancer persister cells [55,56].

We will first define each of these forms of resistance in cancer in general and then give examples of each of the current drug types already discussed above and how they may sooner or later lose their efficacy due to resistance development in one of the forms discussed below.

i.Driver oncogene alterations: Alterations in the target gene, such as mutations and amplifications, allow cells to proliferate despite the presence of inhibitors. Selective pressure from targeted therapies can induce aberrations that reactivate the driver oncogene. Initially, malignancies with these aberrations show substantial responses to selective inhibitors. However, resistance tends to develop over time. Genetic aberrations in the target gene can be categorized as on-target or off-target resistance [23]. On-target effects correspond to enhanced and adverse pharmacological effects specifically at the intended target in the test system. Off-target effects, on the other hand, involve adverse effects due to modulation of other targets that may be biologically related or completely unrelated to the intended target [57]. Off-target resistance involves abnormal activation of alternative signaling pathways, coexisting driver oncogenes, lineage plasticity, epithelial-mesenchymal transition, and persistent cancer cells [58].ii.Downstream or parallel bypass signaling: Three primary oncogenic signaling pathways, namely the PI3K/AKT/mTOR (PI3K pathway), RAS/RAF/ERK (MAPK pathway), and STAT/JAK pathways, regulate cell growth, proliferation, and cellular metabolism as represented in Figure 5 [59].iii.Abnormal signaling activation processes often result from gain-of-function mutations, genomic amplification, chromosomal rearrangements, or autocrine activation. Gain-of-function mutations occur when downstream signaling persists despite upstream blockade by targeted agents. Gene amplification refers to an increase in the number of gene copies [23]. In cancer cells, gene amplification occurs when signals from surrounding cells or the environment induce the production of multiple gene copies [58]. Chromosomal rearrangements can induce cancer either by creating a fusion gene or by disrupting gene regulation [60]. Autocrine signaling refers to the production and release of a signaling molecule by a cell, which then binds to receptors on the same cell to initiate signaling [61].iv.Presence of co-dominant driver oncogenes: It is evident that a number of non-responsive mutations coexist, resulting in resistance to targeted therapies [23].v.Tumor lineage plasticity: Lineage plasticity refers to the ability of a cell to undergo phenotypic transformation toward a different developmental lineage [62]. Cancer cell plasticity arises due to exposure and selective pressure of targeted therapeutic agents [62,63]. This transformation allows tumor cells to adapt to challenging conditions, such as a hypoxic tumor microenvironment [62]. Although the driver mutation is retained, transformed cells no longer rely on it for proliferation, leading to therapeutic resistance. Addressing the epigenetic, genomic, and microenvironmental factors that drive lineage plasticity will be critical to the development of innovative treatment approaches [23].vi.Epithelial-mesenchymal transition: Although the driver mutation is retained, the transformed cells no longer rely on it for proliferation, leading to therapeutic resistance. Addressing the epigenetic, genomic, and microenvironmental factors that drive lineage plasticity will be critical to the development of innovative treatment approaches [58]. It is hypothesized that the process involves the stimulation and activation of intracellular signaling pathways, resulting in the reduction of E-cadherin. EGFR TKIs may induce cells to transition to a mesenchymal phenotype characterized by decreased E-cadherin expression and expression of mesenchymal markers such as N-cadherin [64].vii.Persister cancer cells: The concept of drug-tolerant persister cells has emerged as an important concept. Persister cancer cells refer to a group of cells that can survive systemic treatments by entering a reversible and sluggish proliferative state [65,66] and are thought to be distinct from cancer stem cells. Persister cells typically do not possess conventional driver alterations associated with drug resistance, and their resistant properties may be transient and reversible upon cessation of drug treatment [67]. In contrast, cancer stem cells present in a tumor possess the capacity for self-renewal and the generation of diverse cancer cell lineages [68].viii.The survival of persister cells is attributed to either pre-existing drug-resistant cells or the induction of intrinsic changes that facilitate phenotypic variation [23]. Malignant cells experience various stresses during proliferation, including metabolic, hypoxic, and nutrient limitations. Case reports demonstrate that EGFR inhibitors contribute to chromatin repression, which affects the development and survival of drug-tolerant persister cells [66].

## 4. Resistance Mechanisms Towards the Different Classes of Drugs

We will now look in detail at the mode of resistance in mCRC to the various targeted therapies discussed above. The list of drugs used in the treatment of CRC are listed in Table 2 for reference.

### 4.1. Resistance to Antiangiogenic Therapy (VEGF and VGFR)

Resistance to anti-VEGF therapy, observed in various cancers, occurs due to compensatory activation of alternative signaling pathways and alternate excretion of angiogenesis-related proteins [69]. The upregulation and overexpression of Placental Growth Factor (PIGF) in CRC cases that exhibit resistance to anti-angiogenic therapies indicates the critical role of PIGF as a contributing factor in the development of resistance to anti-VEGF treatments [18].

The angiopoietin/tyrosine kinase with Ig-like and EGF-like domains (TIE) signaling pathway plays a role in blood vessel formation and stability by controlling the activation of downstream pathways. These pathways can be modulated by angiopoietin-2 and elevated levels of angiopoietin-2 have been associated with resistance to bevacizumab treatment [70]. Simultaneous targeting of VEGF and angiopoietin-2 can effectively inhibit the proliferation and progression of cancer cells resistant to therapies specifically targeting VEGF.

Similarly, increased activity of the FGF/FGFR pathway has been linked to anti-VEGF treatment resistance [71,72]. In preclinical studies, simultaneous inhibition of both the FGF/FGFR and VEGF/VEGFR pathways showed promising results against tumor cells. However, in clinical trials, drugs such as nintedanib and the dual blocker of FGF and VEGF, dovitinib, did not show significant benefit in patients refractory to anti-VEGF therapies [73].

The primary mechanism associated with the loss of efficacy of anti-VEGF agents is compensatory activation of the c-MET pathway [74]. However, there is limited evidence regarding the simultaneous targeting of c-MET and VEGF in CRC. Furthermore, a study of combined c-MET and VEGF blockade conducted in NSCLC failed to demonstrate superior effects [75,76]. The results of the study suggest that IL-1 and platelet-derived growth factor receptor are up-, whereas macrophage migration inhibitory factor is downregulated [77,78]. These factors have been observed in several cancer types resistant to VEGF blockade, suggesting a link to resistance to antiangiogenic therapies. However, further confirmation of their utility in CRC therapy is limited due to insufficient clinical data.

### 4.2. Mode of Resistance to EGFR Inhibitors

The most important type of resistance is the presence of mutations in the RAS genes (KRAS, NRAS) in CRC. These mutations can lead to constitutive activation of downstream signaling pathways, bypassing the inhibitory effects of EGFRi and leading to primary resistance [79,80]. As a result of these discoveries, the approvals for these drugs were revised to restrict their use to patients with RAS wild-type disease. RAS mutations remain the most significant factor negatively affecting response to anti-EGFR therapy [81]. In addition to RAS mutations, the presence of BRAFV600E mutations and HER2 amplification have also been shown to have a negative predictive value regarding the efficacy of EGFR inhibitors [82].

Di Nicolantonio and colleagues first conducted a study on a group of 79 mCRC patients who were treated with either panitumumab or cetuximab. Among these patients, the response rate was found to be 0% in the 11 individuals with BRAFV600E-mutant mCRC, while it was 32% in RAS/BRAF wild-type mCRC [83]. The presence of HER2 amplification is associated with significantly worse outcomes in patients receiving EGFR antibodies, with or without cytotoxic chemotherapy. In a retrospective analysis of 74 patients with HER2-amplified mCRC, significantly lower objective response rates (31.2% vs. 46.9%) and shorter median progression-free survival (5.7 vs. 7.0 months) were observed compared to patients without HER2 amplification [84]. The role of both BRAFV600E mutation and HER2 amplification in mCRC is inconclusive due to lack of statistical power. A newly identified group of mutations known as EGFR ectodomain mutations has emerged as a mechanism of secondary resistance to EGFRi. These mutations occur within the receptor region of EGFR and affect the binding of cetuximab or panitumumab [85,86]. Mathematical modeling suggests that patients with mCRC typically harbor numerous subclonal mutations. These mutations, although present, generally have a growth disadvantage compared to the dominant (wild-type) clone. However, targeted therapy disrupts this dynamic by selectively inhibiting the dominant clone, allowing subclonal populations resistant to the inhibitor to grow. When the selective pressure is removed, the subclonal resistance mutations rapidly decline. This interplay between dominant and subclonal populations has been demonstrated in several well-designed preclinical and clinical studies [87,88,89].

In a study of 135 patients with RAS/BRAF wild-type mCRC who progressed on anti-EGFR therapy, researchers determined the half-life of acquired mutations after discontinuation of EGFRi. The half-life of acquired RAS mutations was 3.4 months, while EGFR mutations had a half-life of 6.9 months. In addition, the study demonstrated an approximately 20% improvement in objective response rate for patients who received anti-EGFR therapy within one half-life compared to those who received therapy after more than two half-lives [88].

Even rechallenging with EGFRi seems promising treating resistant CRC. Recent trials showed that EGFR rechallenge is possible and effective [89]. They included patients who initially responded to cetuximab-based chemotherapy, became resistant, and were treated with cetuximab-free regimens subsequently. In the CRICKET trial, cetuximab and irinotecan rechallenge resulted in a 21% objective response rate and a 54% disease control rate. In the CHRONOS trial, panitumumab resulted in a 30% objective response rate and a 63% disease control rate [90,91]. The still ongoing PULSE trial (COLOMATE companion trial) aims to evaluate panitumumab rechallenge in mCRC patients who progressed on prior anti-EGFR therapy based on circulating free DNA profiling [92].

### 4.3. Primary and Secondary Resistance to BRAFV600E Inhibitors

BRAFV600E mutations are present in approximately 7% of mCRC patients and are associated with a markedly inferior response to chemotherapy and an unfavorable prognosis [93,94]. In contrast to patients with BRAFV600E-mutated melanoma, individuals with mCRC generally experience limited or negligible therapeutic benefits when treated with single-agent BRAF inhibitors [93,94]. The primary resistance mechanism involves the reactivation of MAPK signaling [95,96].

Although the combination of EGFR and BRAF inhibitors has demonstrated encouraging response improvements, the durability of these responses is typically constrained by the rapid emergence of secondary resistance. Secondary resistance to BRAF inhibitors develops more rapidly and is more prevalent than in other targeted therapies. The acquisition of RAS mutations or amplifications are observed in approximately 70% of secondary-resistant patients [97]. Various further mechanisms, including BRAF and MET amplifications, BRAF exon 2–8 deletions and EGFR ectodomain mutations have been identified. They have in common, that they all are reactivate MAPK pathway signaling, thereby circumventing the inhibitory effects of the therapy [82].

It was hypothesized that a mutation resistance decline could be expected in BRAFV600E-mutant mCRC under BRAF inhibitory therapy, when the latter is stopped. The BRAFV600E rechallenge arm of the COLOMATE trial currently enrolls patients with BRAFV600E-mutant mCRC who experience secondary resistance to targeted inhibition. These patients will be receiving chemotherapy followed by a rechallenge with encorafenib, cetuximab, and binimetinib. The inclusion of binimetinib in this trial creates negative selective pressure downstream of BRAF, which may reduce the resurgence of secondary resistance mutations that circumvent BRAF signaling [98].

### 4.4. Resistance to HER2 Inhibition

Approximately 3% of patients with mCRC exhibit HER2 amplification [99]. A number of trials have assessed the efficacy of various anti-HER2-based therapies in patients with mCRC, with varying degrees of success. This is largely due to the fact that the majority of patients develop primary or secondary resistance [100,101].

The HERACLES and MyPathway trials have yielded insights into the mechanisms of resistance. In the HERACLES trial, resistance to HER2-targeted therapy was observed as a consequence of the activation of downstream signaling pathways, including the MAPK and PI3K-AKT pathways. Moreover, alterations in the HER2 pathway, including HER2 mutations and amplifications, were identified as potential mechanisms of resistance. The MyPathway trial demonstrated that the activation of alternative signaling pathways, such as the EGFR pathway, can contribute to resistance to HER2 inhibition in mCRC [102,103].

### 4.5. Challenges of ICI in CRC

In mCRC, a notable response was observed in some phase I trials [104]. However, subsequent studies revealed that only a modest proportion of patients with CRC exhibited a response to ICI therapy [105]. It has been demonstrated that only patients with a high tumor mutational burden and whose tumor exhibits elevated levels of MSI-H or dMMR respond to ICI therapy [106,107].

In light of the suboptimal outcomes observed in patients with proficient MMR or MSS CRC, who represent the majority of CRC patients [1], the efficacy of ICI therapy in this population remains a subject of concern. The precise mechanism has yet to be elucidated. Researchers have endeavored to surmount the resistance of MSS CRC to ICI [108]. A number of hypotheses have been proposed to explain this poor response rate, including reduced tumor-specific antigen expression, defects in antigen presentation, alterations in immunosuppressive pathways (e.g., activation of MAPK and loss of PTEN), alternative activation of other immune checkpoint signaling pathways, immune regulatory cells, and cytokines. These hypotheses are currently being investigated [109]. Moreover, strategies to enhance the efficacy of ICI in MSS CRC are currently being developed. These include combined therapy with various approaches, such as radiotherapy, bispecific antibody therapy, other immune checkpoint modulators, and other targeted agents [110].

## 5. Toxicity

It has become evident that a single treatment approach is insufficient for the cure of cancer. Consequently, the combination of targeted therapy with chemotherapy, radiotherapy, or other targeted agents has been demonstrated to be a crucial aspect of an effective treatment plan. However, a potential drawback of combining multiple agents to treat cancer is the increased toxicity that may result. In instances where a single targeting agent is ineffective, the combination of these agents with other anticancer agents, including standard chemotherapy, radiotherapy, or additional drugs targeting distinct tumor-related targets, is a necessary course of action [5,111,112]. However, while the expectation of synergistic anti-tumor effects from combination therapies is high, the potential for synergistic toxicity is also significant, potentially leading to life-threatening complications during clinical trials [5,113]. This phenomenon has been observed in clinical trials involving combinations of two targeted agents, such as sorafenib and bevacizumab. Both agents act on the VEGFR pathway, with one acting intracellularly and the other at the cell surface [114]. The trial was terminated prematurely due to the observation of significant toxicity at the initial dose combination. In the PARASOL trial, the adverse effects of the combined treatment with bevacizumab and pazopanib were found to outweigh the limited effectiveness observed in 22% of patients [112].

### 5.1. Combination Therapy Leads to Higher Toxicity

Combination therapy, which entails the utilization of pharmaceutical agents that target disparate signaling pathways, represents a strategy for addressing both intrinsic and acquired resistance in the context of cancer treatment. However, the chronic oral administration and associated toxicity of targeted therapies present significant limitations to their use in combination, thereby hindering the potential for curative outcomes. Previous efforts to combine targeted therapies that inhibit complementary pathways have been unsuccessful, largely due to the toxicity issues that have been encountered [23].

Initially, targeted agents were presumed to possess minimal toxicity, given that they selectively bind to cancer-related molecules. However, the effects of these agents on both intended and unintended targets can result in toxicity. Such adverse reactions may impede the administration of therapy in accordance with the recommended schedule and dose, thereby potentially compromising the drug’s effectiveness [23]. The specific list of side effects of each drug and drug class for mCRC treatment is shown in Table 3.

### 5.2. On-Target Toxicity

On-target toxicity refers to the inhibition of a specific protein in cells. Initially, targeted agents were presumed to exhibit minimal toxicity, given that their objective was to selectively inhibit cancer-specific targets. However, due to the complex nature of cell signaling and the inability of these agents to exclusively block aberrant proteins in cancer cells, the inhibition of a fixed set of proteins in normal cells can result in toxicity by disrupting signaling pathways. Such on-target toxicities are typically observed across a class of drugs, with drugs within the same class exhibiting comparable toxic effects. For example, inhibition of PI3K has been associated with hyperglycemia, inhibition of vascular endothelial growth factor with hypertension, and inhibition of EGFR with the development of a skin rash [115,116,117,118].

### 5.3. Off-Target Toxicity

Off-target toxicity refers to the phenomenon of a drug inadvertently blocking a non-target protein, which can result in the manifestation of undesirable side effects. In contrast to on-target toxicity, off-target effects do not typically manifest in a class-wide manner and are thought to be shaped by the pharmacological attributes of the specific drug in question. A noteworthy example is the cardiac toxicity associated with osimertinib, which is not observed with gefitinib. Osimertinib, an inhibitor of EGFR in the ErbB/HER family, also demonstrates some activity against the HER2 receptor, which may contribute to its cardiotoxic effects, which may manifest as heart failure, left ventricular dysfunction, conduction abnormalities, and myocardial injury or dysfunction [119].

Drug interactions that result in altered plasma levels of targeted agents may lead to reduced efficacy or increased toxicity [23].

The metabolism of certain targeted agents may be influenced by smoking. A study demonstrated that current smokers exhibited reduced exposure to erlotinib, both at a 150 mg and 300 mg dose, in comparison to non-smokers. The findings indicated that erlotinib is metabolized more rapidly in current smokers [120].

Moreover, factors such as ethnicity, food effect, and age can impact the toxicity, efficacy, and tolerability of drugs [23].

## 6. Cost and Access

Targeted cancer therapies for colorectal malignancies face significant accessibility challenges globally due to high costs. These specialized treatments incur substantial research, development, and regulatory approval expenses, with patent protections often precluding more affordable generic alternatives. Insurance coverage varies widely, leaving many patients to bear prohibitive out-of-pocket costs that render these therapies inaccessible, especially in low- and middle-income countries where economic disparities prevent most from affording them. Additionally, healthcare infrastructure in these regions frequently lacks the capacity for administering complex treatments and conducting necessary genetic testing [121,122].

Access is further constrained by the centralization of treatment facilities in urban centers, leaving rural populations underserved. Regulatory barriers in less developed nations can delay the approval of novel therapies, hindering timely access. Pharmaceutical companies often prioritize markets with higher profit potential, depriving underserved regions of new treatment options. Reliance on imports, exacerbated by tariffs and logistical costs, inflates treatment prices in many low- and middle-income countries. Limited local manufacturing capacity impedes the widespread production of biosimilars, maintaining high costs. Patients may face severe financial hardship from treatment expenses, impacting their adherence and quality of life. This financial burden exacerbates health inequities, as access to these therapies becomes a privilege for the affluent. Without concerted international efforts to promote affordable pricing, build healthcare infrastructure, and facilitate access, targeted therapies risk widening the global gap in cancer care. Collaborative strategies are essential to ensure that advancements in colorectal cancer treatment benefit patients worldwide, not just in wealthier nations [123,124].

## 7. Conclusions and Future Perspective

It is recommended that genomic biomarkers be integrated into routine clinical practice with the goal of improving treatment with targeted agents and making them available for the general population worldwide.

The expansion of tumor genotyping accessibility, exemplified by amplicon-based NGS integrated with copy number, gene fusion, and outlier gene expression panels, represents a pivotal stride in the integration of genomic biomarkers into routine clinical practice. These panels are currently employed in both academically driven clinical research, predominantly as customized panels, and in industry-sponsored trials, typically as commercial panels [124].

Nevertheless, the use of a multigene panel to select a targeted therapy for a patient with mCRC remains contingent upon the availability of a treatment that targets the specific alteration in question [125,126].

Still, the utilization of circulating cell-free tumor DNA (ctDNA) as the principal source of material for NGS has, the potential to enhance the enrollment rate in clinical trials while maintaining the efficacy of treatment when compared to tissue genotyping [119].

The analysis of circulating tumor DNA (ctDNA) obtained from liquid biopsies has been proposed as a method for improving tumor genotyping. Liquid biopsy sampling is minimally invasive and can be repeated several times, thus overcoming the spatial and temporal heterogeneity issues associated with tissue biopsy samples [127]. Initially, ctDNA analysis was employed to optimize treatment with anti-EGFR antibodies in the metastatic setting. However, the advent of next-generation sequencing has enabled the identification of most genomic aberrations in ctDNA, greatly extending the investigative potential of this approach [119]. The analysis of ctDNA has facilitated the enhanced identification of resistance mutations, and serial monitoring of ctDNA has also been employed to assess responses to therapy [120]. Despite the recognition of the analytical validity and clinical utility of this approach [110], several hurdles, including cost-effectiveness and the optimization of the pre-analytical steps, limit the implementation of ctDNA-based analysis in routine clinical practice and in CRC management guidelines [128].

The current standard procedure for assessing spatial heterogeneity is the use of whole-exome sequencing or high-coverage multigene panels carried out on multiple biopsy sites. Nevertheless, ultra-deep sequencing may be necessary for an accurate assessment of intratumor heterogeneity. Digital PCR also has a high sensitivity, but is costly and presents certain technical challenges in the context of these kinds of studies [129]. In addition to omic studies aimed at further characterizing CRC, more targeted approaches could be employed to enhance patient management. Recently, NGS targeted panels have been proposed for use in clinical practice, as they are less expensive and time-consuming than other methods [128,130]. Nevertheless, their utility remains a topic of debate, and the genetic alterations confined to the RAS and BRAF mutation spectrum remain the most extensively investigated markers [37].

Furthermore, the investigation of the mechanisms underlying toxicity has been markedly underrepresented. A multitude of toxicities have been identified during clinical trials, yet the emphasis has predominantly been on the provision of supportive care for patients, with minimal attention directed towards fundamental research elucidating the mechanisms underlying these toxicities. The majority of treatment guidelines have been formulated based on those established for the known adverse effects induced by chemotherapy, rather than on guidelines tailored to the specific agents utilized in targeted therapy. It is imperative that there be an increased emphasis on the generation of fundamental scientific evidence in order to establish guidelines that are suitable for targeted therapies. Further research is essential to address the questions surrounding the prevention of toxicity without compromising the antitumor effect, the effective management of the complex combined adverse effects associated with combination therapy, and the understanding of the mechanistic distinctions between the toxicities of cytotoxic and targeted treatments [5].

Future investigations should prioritize the elucidation of toxicity mechanisms and the development of innovative drug administration schedules that enable the combination of TKIs targeting complementary pathways. Furthermore, studies should address the impact of special populations and host factors on variations in toxicity. The issue of the unaffordability of targeted agents for the majority of cancer patients globally is a matter of significant concern. It is imperative that urgent global initiatives be implemented to mitigate the exorbitant prices of pharmaceuticals [23].

Overcoming mCRC treatment challenges is of major clinical importance, since it directly affects patient survival rates, quality of life, and the overall burden of the disease. mCRC remains a major cause of cancer related mortality worldwide, despite the advancements in chemotherapy, targeted therapies, and immunotherapy [131]. Improved prognosis and prolonged survival can be reached through effective treatment strategies, which include overcoming drug resistance, enhancing tumor response, and optimizing personalized medicine approaches. Additionally, the development of novel therapeutic agents and biomarker-driven treatments could help identify patients who are most likely to benefit from specific interventions and hence minimizing toxicity and maximizing efficacy [132]. Continued research into overcoming treatment resistance and metastatic progression is crucial for transforming mCRC from a fatal disease into a manageable one.

## Figures and Tables

**Figure 1 cancers-17-01098-f001:**
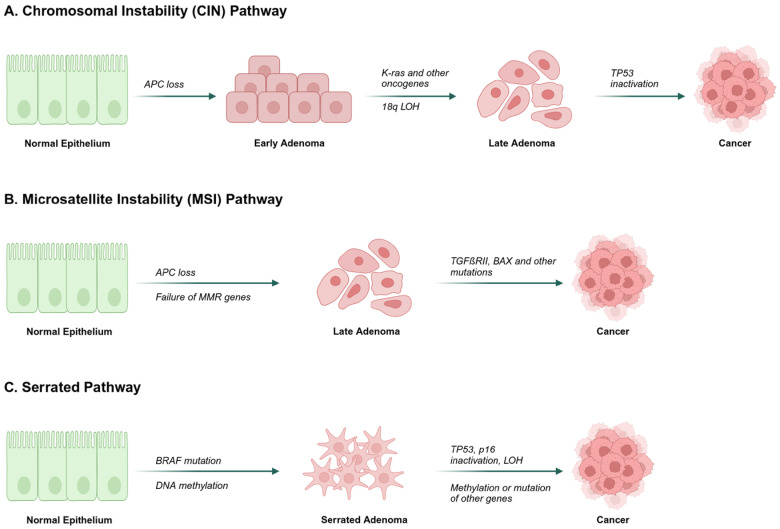
Pathogenesis of CRC. This scheme illustrates the progression of CRC via three distinct molecular pathways: chromosomal instability (characterized by APC loss, KRAS mutation, and p53 inactivation), microsatellite instability (due to mismatch repair deficiencies), and the serrated pathway (involving BRAF mutation). This concise depiction encompasses the pivotal genetic and epigenetic alterations that orchestrate the transformation from normal colonic epithelium to invasive carcinoma.

**Figure 2 cancers-17-01098-f002:**
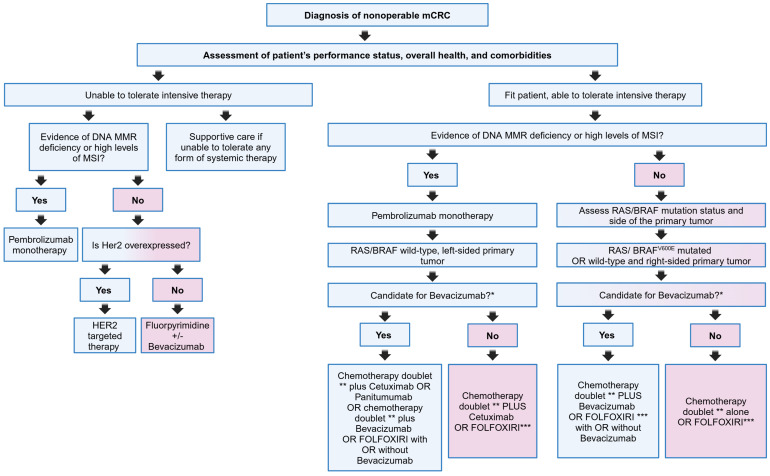
Flow diagram describing the approach to later lines of systemic therapy in mCRC. The following has been in the original Table and seems to got lost during the revision process: HER2: human epidermal growth factor receptor 2; FOLFOX: oxaliplatin plus leucovorin and short-term infusional fluorouracil; FOLFOXIRI: infusional fluorouracil, leucovorin, oxaliplatin, and irinotecan; FOLFIRI: irinotecan plus leucovorin and short-term infusional fluorouracil; * Contraindications to bevacizumab include: Major surgery within 28 days, active bleeding, and untreated hemorrhagic brain metastases; ** FOLFOX or FOLFIRI are Backbone chemotherapy doublets appropriate for use with cetuximab or panitumumab. Oxaliplatin plus capecitabine (CAPOX/XELOX) is another doublet that can be used alone or in combination with bevacizumab. Patients who received adjuvant oxaliplatin-based chemotherapy within the last 12 months would benefit more from FOLFIRI that from an oxaliplatin-containing regimen; *** Triplet therapy, although more toxic than doublet therapy, is preferred for patients with a good PS who are able to tolerate it and who have biologically aggressive/poor-prognosis cancer.

**Figure 3 cancers-17-01098-f003:**
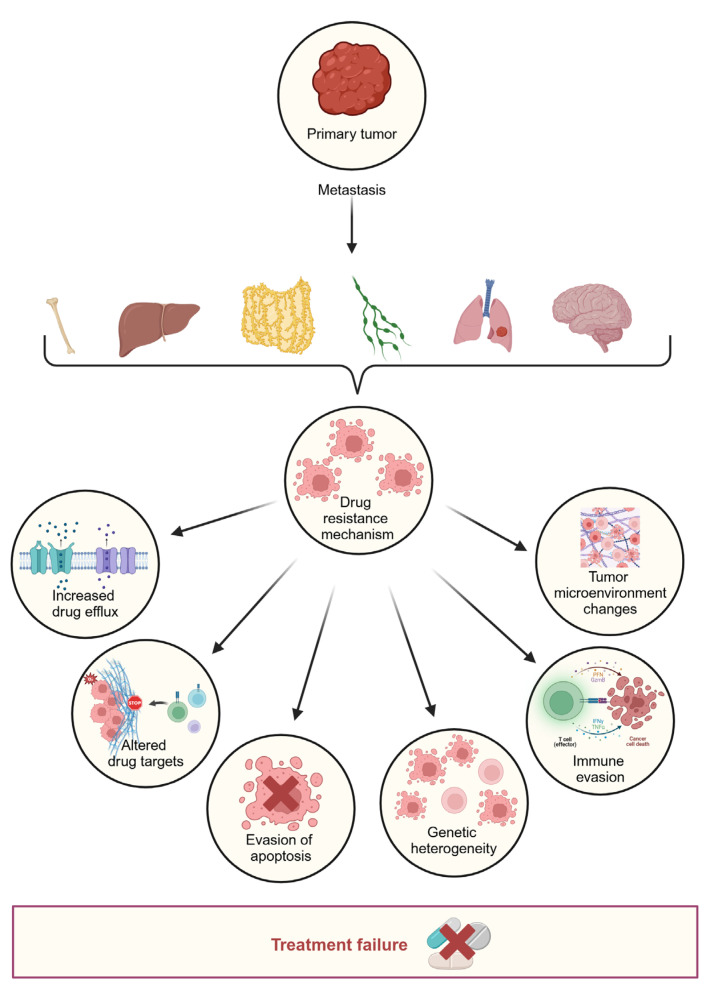
How metastasis leads to drug resistance. The process of metastasis-driven progression to distant organs promotes diverse resistance strategies, including immune modulation, drug efflux, and microenvironmental adaptations. Ultimately, these factors can contribute to therapeutic failure.

**Figure 4 cancers-17-01098-f004:**
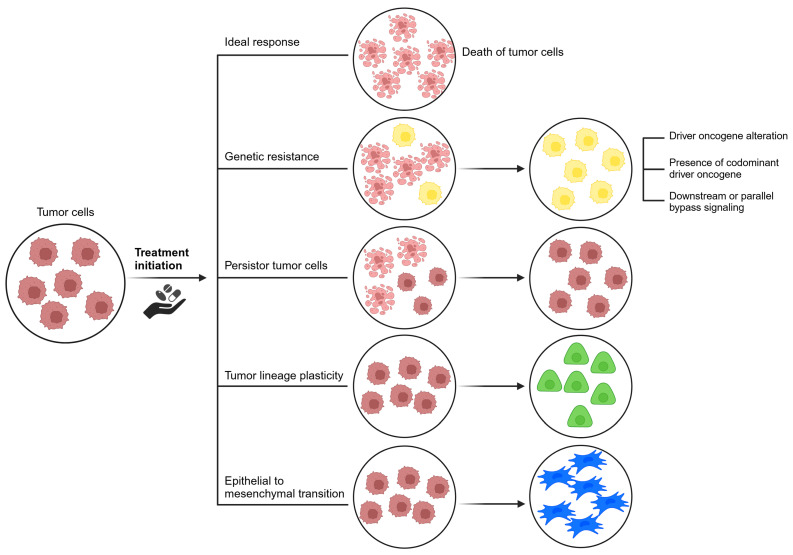
Different modes of acquired resistance.

**Figure 5 cancers-17-01098-f005:**
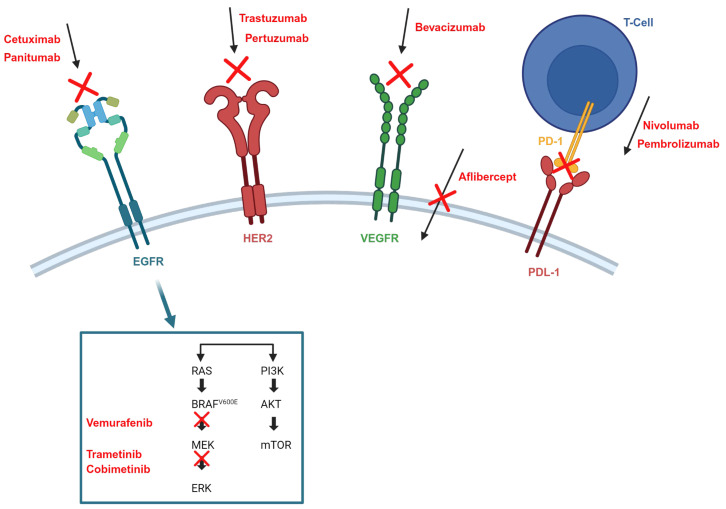
Different signaling pathways involved in cancer development and interfering drugs.

**Table 1 cancers-17-01098-t001:** Summary of clinical outcomes for targeted therapies in mCRC.

Target	Drug(s)	Clinical Indication	Key Clinical Outcomes	Clinical Observations
EGFR	Cetuximab, Panitumumab	RAS wild-type; first-/second-line or post chemotherapy	ORR of ~40–50%; reduced effectiveness in right-sided tumors	Benefit is confined to RAS wild-type tumors; intrinsic resistance is common in right-sided cancers
BRAF V600E	Encorafenib (+Cetuximab ± Binimetinib)	BRAF V600E-mutant; typically second-line	Improved OS (9.0–9.3 months vs. 5.4 months with SOC); ORR of 20–26% versus 2% with SOC	Median survival around 12 months
HER2	Trastuzumab ± Pertuzumab	HER2-positive; second-line	ORR 9.7% to 35%, with some reports up to 45% using trastuzumab; potential pulmonary toxicity	Dual HER2 blockade more effective than single
MEK	Trametinib, Cobimetinib	Later lines, typically in combination	Limited activity as monotherapy; enhanced outcomes in combination therapies	Not approved as monotherapy; benefits are mainly observed in combination regimens
Immune Checkpoints	Pembrolizumab, Nivolumab (± Ipilimumab)	MSI-H/dMMR	Marked improvements in response and survival in MSI-H/dMMR tumors; robust benefit in selected patient groups	Minimal benefit in microsatellite stable mCRC; efficacy is confined to MSI-H/dMMR subgroups

SOC: standard of care.

**Table 2 cancers-17-01098-t002:** List of drugs used in the therapy of mCRC.

Mechanism of Action	Medication
VEGF Inhibitor	Bevacizumab (Avastin^®^)
	Aflibercept (Eylea^®^, Zaltrap^®^)
EGFR inhibitor	Cetuximab (Erbitux^®^)
	Panitumumab (Vectibix^®^)
BRAF Inhibitor	Vemurafenib (Zelboraf^®^)
	Encorafenib (Braftovi^®^)
HER2 Inhibitor	Pertuzumab (Perjeta^®^)
	Trastuzumab (Herceptin^®^)
MEK1, MEK2 Inhibitor	Trametinib (Mekinist^®^)
MEK 1 Inhibitor	Cobimetinib (Cotellic^®^)
PD-1 Inhibitor	Pembrolizumab (Keytruda^®^)
	Volumab (Opdivo^®^)

**Table 3 cancers-17-01098-t003:** List of side effects of each drug and drug class for mCRC treatment discussed in the first part of the review.

Medication	Brand Name	Mechanism of Action	Mechanism of toxicity
Bevacizumab	Avastin^®^	VEGF inhibitor	1. Cardiovascular Toxicity: VEGF inhibitors can affect the cardiovascular system by disrupting the normal balance of blood vessel growth and maintenance. By inhibiting VEGF, these medications can lead to hypertension and increase the risk of cardiovascular events such as heart attack or heart failure.
Aflibercept	Eylea^®^, Zaltrap^®^	VEGF Inhibitor	2. Impaired Wound Healing: VEGF plays a crucial role in the process of wound healing, as it promotes angiogenesis and the growth of new blood vessels in injured tissues. By inhibiting VEGF, VEGF inhibitors can impair wound healing and increase the risk of delayed healing or complications following surgery.
			3. Hemorrhage: VEGF inhibitors can interfere with the formation and maintenance of blood vessels, leading to fragile blood vessels that are more prone to bleeding. This can result in an increased risk of hemorrhage, both externally and internally.
			4. Proteinuria: VEGF inhibitors can affect the filtration function of the kidneys, leading to an increased excretion of protein in the urine. This occurs due to the disruption of normal blood vessel development and maintenance in the kidneys.
			5. Gastrointestinal Perforation:. The inhibition of VEGF can weaken the integrity of the gastrointestinal tract, leading to the development of holes or tears in the stomach, intestines, or other parts of the gastrointestinal system.
Cetuximab	Erbitux^®^	Targets EGFR	1. Skin Toxicity: EGFR inhibitors can cause skin-related toxicities, such as rash and dermatitis. This occurs because EGFR is also expressed in the skin, and inhibiting EGFR can disrupt normal skin cell growth and maintenance.
Panitumumab	Vectibix^®^	Targets EGFR	2. Diarrhea: EGFR inhibitors can affect the gastrointestinal tract and lead to an increased frequency of bowel movements and diarrhea. The exact mechanism of EGFR inhibitor-induced diarrhea is not fully understood, but it is thought to be related to the effects of EGFR inhibition on the gut lining and water absorption.
			3. Nail Changes: EGFR inhibitors may cause changes in the nails, including nail discoloration, brittle nails, or nail inflammation (paronychia). These nail changes are generally reversible once the treatment is completed or the dose is adjusted.
			4. Mucositis: EGFR inhibitors can result in inflammation and ulceration of the mucous membranes, leading to mucositis. This can affect the lining of the mouth, throat, and gastrointestinal tract, causing pain, difficulty swallowing, and mouth sores.
			5. Ocular Toxicity: EGFR inhibitors may cause ocular toxicities such as dry eyes, conjunctivitis (inflammation of the conjunctiva), and corneal erosion.
Vemurafenib	Zelboraf^®^	BRAF Inhibitor	1. Cutaneous Toxicity: BRAF inhibitors can cause skin-related toxicities, including rash, photosensitivity, and hyperkeratosis (thickening of the outer layer of the skin). These skin toxicities can manifest as dryness, redness, itching, or the development of acneiform eruptions. Regular monitoring of the skin and appropriate skincare measures are important to manage these toxicities.
Encorafenib	Braftovi^®^	BRAF Inhibitor	2. Pyrexia (Fever): Fever is a common side effect associated with BRAF inhibitors. It is usually low-grade and self-limiting, but occasionally it can be severe and require medical attention.
			3. Gastrointestinal Toxicity: BRAF inhibitors can cause gastrointestinal toxicities, such as diarrhea and nausea. Diarrhea can range from mild to severe and may require supportive care and management to prevent dehydration. Nausea and vomiting can also occur.
			4. Hepatotoxicity: There have been reports of liver toxicity associated with BRAF inhibitors, including elevation of liver enzymes.
			5. Cardiotoxicity: In some cases, BRAF inhibitors have been associated with cardiotoxic effects, including arrhythmias, left ventricular dysfunction, and cardiomyopathy.
			6. Photosensitivity: BRAF inhibitors can make the skin more sensitive to sunlight, leading to an increased risk of sunburn.
Trastuzumab	Herceptin^®^	HER2 Inhibitor	1. Cardiotoxicity: HER2 inhibitors can have cardiotoxic effects, including a risk of decreased heart function and heart failure. This occurs because HER2 plays a role in the normal functioning and maintenance of heart cells.
Pertuzumab	Perjeta^®^	HER2 Inhibitor	2. Infusion Reactions: HER2 inhibitors are typically administered intravenously, and infusion reactions may occur during or shortly after administration. These reactions can include symptoms such as fever, chills, skin rash, itching, shortness of breath, or low blood pressure.
			3. Diarrhea: HER2 inhibitors can cause gastrointestinal toxicities, with diarrhea being a common side effect. The severity of diarrhea can range from mild to severe.
			4. Fatigue: Fatigue or excessive tiredness is a frequent side effect associated with HER2 inhibitors. It can affect a patient’s daily activities and quality of life.
			5. Hepatotoxicity: In rare cases, HER2 inhibitors may cause liver toxicities, such as elevated liver enzymes.
			6. Skin and Nail Toxicities: HER2 inhibitors can lead to skin-related toxicities, including rash, dry skin, and changes in the nails. Some patients may experience skin redness, itching, or skin peeling. Nail changes, such as discoloration or brittleness, can also occur.
Trametinib	Mekinist^®^	MEK1, MEK2 inhibitor	1. Dermatological Toxicity: MEK1 inhibitors can cause various dermatological toxicities, including rash, acneiform eruptions, dry skin, and pruritus (itching). These skin-related toxicities are commonly observed and can vary in severity.
Cobimetinib	Cotellic^®^	MEK1 inhibitor	2. Gastrointestinal Toxicity: MEK1 inhibitors can cause gastrointestinal toxicities, such as diarrhea, nausea, vomiting, and abdominal pain. Diarrhea is a common side effect and can range from mild to severe.
			3. Hepatotoxicity: MEK1 inhibitors have been associated with hepatotoxic effects, including elevation of liver enzymes (transaminases) and hepatocellular injury.
			4. Ocular Toxicity: MEK1 inhibitors can cause ocular toxicities, including dry eyes, blurred vision, and ocular inflammation.
			5. Cardiovascular Toxicity: In some cases, MEK1 inhibitors may lead to cardiovascular toxicities, including cardiomyopathy and prolongation of the QT interval.
			6. Interstitial Lung Disease: Rarely, MEK1 inhibitors have been associated with interstitial lung disease, which is characterized by inflammation and scarring of lung tissue. Symptoms may include shortness of breath, cough, and fever.
Pembrolizumab	Keytruda^®^	PD-1 Inhibitor	1. Immune-Related Adverse Events: Anti-PD-1 inhibitors can cause immune-related adverse events, which occur due to the activation of the immune system. These adverse events can affect various organs and systems in the body. Common irAEs are listed subsequently.
Nivolumab	Opdivo^®^	PD-1 Inhibitor	2. Skin Toxicity: Skin toxicities can include rash, itching, and blistering. More severe reactions such as Stevens-Johnson syndrome or toxic epidermal necrolysis can occur but are rare.
			3. Gastrointestinal Toxicity: Gastrointestinal toxicities can manifest as diarrhea, colitis, or hepatitis. Symptoms may include abdominal pain, diarrhea with or without blood, or jaundice.
			4. Endocrine Toxicity: Endocrine toxicities can result in the dysfunction of various glands in the body, such as the thyroid, pituitary, or adrenal glands. This can lead to conditions like hypothyroidism, hyperthyroidism, adrenal insufficiency, or hypophysitis.
			5. Pneumonitis: Pneumonitis is inflammation of the lungs, which can cause symptoms such as cough, shortness of breath, and chest pain.
			6. Nephritis: Nephritis refers to inflammation of the kidneys, which can cause kidney dysfunction and abnormal urine tests.
			7. Fatigue: Fatigue or excessive tiredness is a common side effect associated with anti-PD-1 inhibitors. It can impact a patient’s daily activities and quality of life.
			8. Infusion Reactions: Infusion reactions may occur during or shortly after the administration of anti-PD-1 inhibitors. These reactions can include symptoms such as fever, chills, itching, rash, or low blood pressure.
			9. Autoimmune Disorders: Anti-PD-1 inhibitors can trigger or exacerbate pre-existing autoimmune disorders or lead to the development of new autoimmune conditions. These can include conditions like rheumatoid arthritis, autoimmune thyroiditis, or type 1 diabetes.

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
