# Peer review of "Addressing Challenges in Targeted Therapy for Metastatic Colorectal Cancer"

_cancers, 2025, doi:10.3390/cancers17071098_

Round 1

Reviewer 1 Report (Previous Reviewer 1)

Comments and Suggestions for Authors

The spelling of BRAFV vs. BRAF V (line 110 et al.) is still varying! 

Author Response

Spelling of BRAFV600E

We apologize for the still incomplete checking, did it again and had it also re-checked by an AI program. It is now BRAFV600E everywhere in the text.

Reviewer 2 Report (Previous Reviewer 2)

Comments and Suggestions for Authors

Overall, this manuscript offers a comprehensive review of targeted therapies for metastatic colorectal cancer (mCRC), addressing key challenges such as intrinsic and acquired drug resistance, tumor heterogeneity, toxicity, and cost issues. The authors have successfully synthesized a vast amount of current literature and presented a detailed overview that integrates both molecular mechanisms and clinical outcomes, making the review highly informative for both researchers and clinicians.

One of the major strengths of the paper is its systematic organization. The discussion is clearly segmented into critical areas, including the mechanisms underlying primary and secondary resistance, as well as the adverse effects associated with various targeted agents. The inclusion of detailed tables that summarize clinical outcomes, drug mechanisms, and toxicity profiles further enhances the utility of the review. Moreover, the exploration of emerging strategies, such as the integration of genomic biomarkers and liquid biopsy techniques, adds a forward-looking perspective that is both timely and relevant.

However, there are several areas that could benefit from refinement. The sections discussing tumor heterogeneity and resistance mechanisms, while thorough, could be made more concise to improve clarity and readability. Some technical details may overwhelm readers who are less familiar with the intricacies of molecular oncology. It would be beneficial to streamline these parts and focus on the most impactful findings. Additionally, the cost and accessibility discussion needs to be expanded, incorporating recent economic data and a more balanced analysis of healthcare disparities across different regions.

In conclusion, this review is well-researched and provides valuable insights into the current challenges of targeted therapy in mCRC. With minor revisions to enhance clarity and update certain sections, the manuscript will serve as a significant resource in the field, offering practical guidance for future research and clinical practice.

Author Response

In general: thank you very much for your constructive feedback on our manuscript. We appreciate the opportunity to clarify the novelty and significance of our work and to address the concerns you raised.

However, there are several areas that could benefit from refinement. The sections discussing tumor heterogeneity and resistance mechanisms, while thorough, could be made more concise to improve clarity and readability. Some technical details may overwhelm readers who are less familiar with the intricacies of molecular oncology. It would be beneficial to streamline these parts and focus on the most impactful findings.

As requested, we removed some of the molecular, technical details. Most have, of course, been untouched since targeted therapies base on certain molecular changes. Still, we feel that it reads now easier.

Additionally, the cost and accessibility discussion needs to be expanded, incorporating recent economic data and a more balanced analysis of healthcare disparities across different regions.

We concur with the assessment of the reviewer that this facet is accruing heightened significance in the context of the emergence of novel targeted therapeutic interventions, the expanding global population, and contemporary political developments. On the other hand and after internal discussion, we decided to stick to the short and condensed version of our cost and accessibility discussion, since the topic is better dealt with in more specific review articles. 

This manuscript is a resubmission of an earlier submission. The following is a list of the peer review reports and author responses from that submission.

Round 1

Reviewer 1 Report

Comments and Suggestions for Authors

 The paper offers a good review on methodolgies for targeted therapy.

Being a clinician I would be interested in the potential and the real response rates achievable in the metastatic target types (which you do not mention, only in p5-6).

The citations need corrections. The text format needs to be homogenous.

There are several remarks to be corrected (indicated by the numbers of line in the  manuscript:

l13 "between patients": In l24 you have a clear expression

17: "capture" DNA: better word!

36: Sometimes you use capital letters in the beginning, somtetimes not. Why?

54: Citations 1, 3, and 4 are the same papers!!!

73: Please give an hint in the text to Figure 1 and in the following to all Fig´s and Tab´s.

80: Ref´s 8-12 appear nowhere in your text! (... or in l 120). Please stick to sequences concerning your ref´s.

92: Is it BRAF V600E (l53) or BRAFV 600E (l 92)

139: You often are citing [24]: The paper´s topic is NSCLC - correct?

223: You switch from cit.51 to 60; what about 52-59?

255: Waht about cit. 65?

285: Text format is switching: Why?

 316: Cit 81 relates to thyroid cancer, not  metastatic colon- or rectal cancer ("colorectal cancer" is an old expressoion.....)

from 360-372: cit´s 97-102 are missing!! I cannot understand how you did these mistakes!!)

383: Is this citation adequate? It dates back to 1987 (your topic at this time was not very present in research....?)

419: Is cit 3 correct or 1 or 4?

From 429: cit´s 118-120 and 123 relate to "liquid biopies - not to toxicities. 121 also?

461: In a table similar to table 1 you might summarize the clinical results, which you would like to know regarding the topic of your paper. 

500: are [29, 30] correct citations?

514: Do you mean cit´s 131, 132 instead of 31,32? ..... mistakes getting on the nerves of a reviewer!!

535: Cit 117 o.k.?

600: "et al." tolerated?

Comments on the Quality of English Language

Some expressions are wrong; it would be good, if language is corrected.

Reviewer 2 Report

Comments and Suggestions for Authors

This article provides an overview of the challenges in targeted therapy for metastatic colorectal cancer, a relatively outdated topic lacking in attractiveness and novelty. The main body of the text primarily focuses on drug resistance mechanisms and concludes with a discussion on liquid biopsy and molecular markers, lacking logical coherence.

Reviewer 3 Report

Comments and Suggestions for Authors

Comment 1: In the introduction section must add a diagram regarding Pathogenesis of CRC

Comment 2: Describe how metastasis leads to drug resistance? With diagram

Comment 3: Drugs used in targeted therapy must be represented in tabular form for better understanding to the readers

Comment 4: How signaling pathways, gut-microbiome, hormonal disbalance, ageing, stress are involved in drug resistance of CRC?

Comment 5: Write down a paragraph on clinical significance of overcoming metastatic CRC treatment

Comment 6: Follow this paper for better improvement: https://doi.org/10.1016/j.adcanc.2024.100114,  

https://doi.org/10.1002/cam4.6502, https://doi.org/10.1007/s12602-023-10183-2

Comments on the Quality of English Language

Ok